# Alcohol Septal Ablation versus Septal Myectomy Treatment of Obstructive Hypertrophic Cardiomyopathy: A Systematic Review and Meta-Analysis

**DOI:** 10.3390/jcm9103062

**Published:** 2020-09-23

**Authors:** Ibadete Bytyçi, Stefano Nistri, Stellan Mörner, Michael Y. Henein

**Affiliations:** 1Institute of Public Health and Clinical Medicine, Umeå University, 90187 Umeå, Sweden; i.bytyci@hotmail.com (I.B.); Stellan.Morner@umu.se (S.M.); 2Universi College, Bardhosh, 10000 Prishtina, Kosovo; 3Clinic of Cardiology, University Clinical Centre of Kosovo, 10000 Prishtina, Kosovo; 4Cardiology Service, CMSR-Veneto Medica, 204-36077 Altavilla Vicentina, Italy; stefanonistri41@gmail.com; 5Molecular and Clinic Research Institute, St George University, Brunel University, London SW17 0QT, UK; 6Institute of Fluid Dynamics, Brunel University, London UB8 3PH, UK

**Keywords:** hypertrophic obstructive cardiomyopathy, alcohol septal ablation, septal myectomy

## Abstract

Surgical myectomy (SM) and alcohol septal ablation (ASA) are two invasive therapies for symptomatic patients with hypertrophic obstructive cardiomyopathy (HOCM), despite medical therapy. This meta-analysis aims to compare the efficacy of the two procedures. We searched all electronic databases until February 2020 for clinical trials and cohorts comparing clinical outcomes of ASA and SM treatment of patients with HOCM. The primary endpoint was all-cause mortality, cardiovascular (CV) mortality, sudden cardiac death (SCD), re-intervention, and complications. Secondary endpoints included relief of clinical symptoms and drop of left ventricular outflow tract (LVOT) gradient. Twenty studies (4547 patients; 2 CTs and 18 cohorts) comparing ASA vs. SM with a mean follow-up of 47 ± 28.7 months were included. Long term (8.72 vs. 7.84%, *p* = 0.42) and short term (1.12 vs. 1.27%, *p* = 0.93) all-cause mortality, CV mortality (2.48 vs. 3.66%, *p* = 0.26), SCD (1.78 vs. 0.76%, *p* = 0.20) and stroke (0.36 vs. 1.01%, *p* = 0.64) were not different between procedures. ASA was associated with lower peri-procedural complications (5.57 vs. 10.5%, *p* = 0.04) but higher rate of re-interventions (10.1 vs. 0.27%; *p* < 0.001) and pacemaker dependency (12.4 vs. 4.31%, *p* = 0.0004) compared to SM. ASA resulted in less reduction in LVOT gradient (−47.8 vs. −58.4 mmHg, *p* = 0.01) and less improvement of clinical symptoms compared to SM (New York Heart Association (NYHA) class III/IV, 82.4 vs. 94.5%, *p* < 0.001, angina 53.2 vs. 84.2%, *p* = 0.02). Thus, ASA and SM treatment of HOCM carry a similar risk of mortality. Peri-procedural complications are less in alcohol ablation but re-intervention and pacemaker implantations are more common. These results might impact the procedure choice in individual patients, for the best clinical outcome.

## 1. Introduction

Hypertrophic cardiomyopathy (HCM) is the most common inheritable heart disease characterized by left ventricular (LV) hypertrophy, diverse clinical presentation and hemodynamic abnormalities [1]. Left ventricular outflow tract obstruction (LVOTO) is a characteristic feature of many patients with HCM [2]. Significant LVOTO is associated not only with symptoms such as chest pain, dyspnea and fatigue, but also with increased risk for all-cause mortality, cardiovascular (CV) mortality, sudden cardiac death (SCD) and other CV complications [3,4,5]. To control symptoms in those patients with hypertrophic obstructive cardiomyopathy (HOCM) when medical therapies fail, septal myectomy (SM) and alcohol septal ablation (ASA), are two invasive treatments that are used, but with no consensus. 

SM has been considered as the gold standard for ventricular septal thickness reduction for more than half a century [6], but ASA has started gaining clinical popularity since the early 1990s [6,7], particularly for patients in whom surgery is contraindicated, considered at high risk, or in patients who have declined surgery [8]. Despite many studies comparing the outcome of the two treatment strategies, debates on the effectiveness of the two methods [9,10,11] still exist, which make the best treatment choice difficult at times. The aim of this meta-analysis is to compare the efficacy of the two treatment procedures on short and long-term clinical outcomes.

## 2. Methods

We followed the 2009 guidelines of preferred reporting items for systematic reviews and meta-analysis (PRISMA) statement [12], which is an amendment to the Quality of Reporting of Meta-analyses (QUOROM) statement [13]. Due to the study design (meta-analysis), neither Institutional Review Board (IRB) approval nor informed patient consent was needed. 

### 2.1. Search Strategy

We systematically searched PubMed-Medline, EMBASE, Scopus, Google Scholar, the Cochrane Central Registry of Controlled Trials and ClinicalTrial.gov, up to March 2020, using the following key words: “Hypertrophic cardiomyopathy” OR “HCM” AND “ Surgical septal myectomy” OR “SM” OR “alcohol septal ablation” OR “ASA”AND “Outcomes” OR “Cardiovascular outcomes” OR “Mortality” “Cardiovascular mortality” OR “Sudden cardiac death” OR “Cardiovascular complications” OR “Left ventricular outflow tract obstruction” OR “LVOTO”.

Additional searches for potential trials included the references of review articles on the subject, and the abstracts from the following congresses: scientific sessions of the European Society of Cardiology (ESC), the American Heart Association (AHA), American College of Cardiology (ACC), and European Association of Cardiovascular Imaging (EACVI). The wild-card term ‘‘*’’ was used to enhance the sensitivity of the search strategy. The literature search was limited to articles published in English and to human studies. No filters were applied. Two reviewers (IB and MYH) independently and separately evaluated each article. The remaining articles were obtained in full-text and assessed by the same two researchers.

### 2.2. Study Selection

Inclusion criteria were: (1) data on two arms, (2) reporting short and/or long-term outcome, (3) follow-up, and (4) enrolled population of adults aged ≥18 years.

Exclusion criteria were: (1) only one group of treatment, (2) insufficient statistical data to compare two groups, (3) no follow-up, (4) studies not in humans and (5) articles not published in English.

### 2.3. Outcome Variables

The primary endpoint was long-term all-cause mortality (during follow-up), short-term all-cause mortality (within 30 days of procedure), cardiovascular mortality (CV), sudden cardiac death (SCD), re-intervention, stroke and peri-procedural complications (tamponade, cerebrovascular accident, ventricular septal defect, occlusion or dissection of coronary arteries, nonfatal cardiac arrest and urgent thoracotomy). Secondary endpoints included improvement of clinical symptoms and fall of LVOT gradient. All endpoints were evaluated at the longest available follow-up according to individual protocols.

### 2.4. Data Extraction 

Eligible studies were reviewed and the following data were abstracted: (1) first author’s name; (2) year of publication; (3) study design; (4) data on two arms; SM and ASA; (5) patients’ baseline characteristics; (7) mean follow-up period; and (9) age and gender of participants.

### 2.5. Quality Assessment

Assessment of risk of bias of clinical trials (CTs) was evaluated by the same investigators using the Cochrane risk of bias. Evaluated items were: random sequence generation, allocation sequence concealment, blinding of participants and personnel, blinding of outcome assessment, incomplete outcome data, selective outcome reporting and other potential sources of bias. The risk of bias in each study was judged to be “low”, “high” or “unclear” [12]. For the assessment of risk of bias in cohort studies we used the Newcastle-Ottawa Scale (NOS). The risk of bias in each study was judged to be “good”, “fair” or “poor” [14].

### 2.6. Statistical Analysis

The meta-analysis was conducted using Statistical analysis, performed using the RevMan (Review Manager (RevMan) Version 5.1, The Cochrane Collaboration, Copenhagen, Denmark), with two-tailed *p* < 0.05 considered as significant. Relative risk (RR) ratios with 95% confidence interval (CI) are presented as summary statistics, whereas for the continuous variable, weighted mean differences (WMD) and 95% CI were used. The baseline characteristics are reported in median and range. Mean and standard deviation (SD) values were estimated using the method described by Hozo et al. [15]. Analysis is presented in forest plots, the standard way for illustrating the results of individual studies and meta-analysis. The meta-analyses were performed with the random-effects model. Heterogeneity between studies was evaluated using Cochrane Q test and *I^2^* index. As a guide, *I^2^* < 25% indicated low, 25–50% moderate and >50% high heterogeneity [16]. To assess the additive (between-study) component of variance, the reduced maximum likelihood method (*tau^2^*) incorporated the occurrence of residual heterogeneity into the analysis [17]. Publication bias was assessed using visual inspections of funnel plots and Egger’s test. 

## 3. Results

### 3.1. Search Results and Trial Flow

Of 7638 articles identified in the initial searches, 645 studies were considered as potentially relevant. After a stringent selection process, 20 articles met the inclusion criteria [18,19,20,21,22,23,24,25,26,27,28,29,30,31,32,33,34,35,36,37]. Two of them were CTs [31,33] and 18 cohort studies [18,19,20,21,22,23,24,25,26,27,28,29,30,32,34,35,36,37]. (Appendix A).

### 3.2. Characteristics of Included Studies

Twenty studies with a total of 4547 patients, 1861 in the ASA group and 2686 in the SM group with a mean follow-up duration of 47 ± 28.7 months were included (Appendix A). Patients treated with ASA were older (54.5 ± 15.1 vs. 48.2 ± 14.6 years, *p* < 0.001) and had a shorter hospital stay (5.1 ± 2.7 vs. 7.8 ± 4.6 days, *p* = 0.02) compared to patients who underwent SM. The female gender distribution was not different in the two patient groups (49 vs. 46.1%, *p* = 0.10, respectively, Appendix A).

### 3.3. Baseline LV and LA Structure and Function in ASA vs. SM

The two groups of patients had similar baseline LV dimensions and function: LV end-diastolic dimension (LVEDD), interventricular septal thickness in diastole (IVSd), and LV ejection fraction (LVEF) (*p* = 0.13, *p* = 0.07 and *p* = 0.78, respectively, Appendix A). The LVOT gradient was not different between the two patient groups (WMD = 2.37 (95% CI, −1.94 to 6.68, *p* = 0.28, *I*^2^ = 58%) and neither was the prevalence of significant (moderate to severe) mitral regurgitation (MR) (*p* = 0.47), whereas the LA size was larger in the SM group as compared to the ASA group (*p* = 0.006, Appendix A).

### 3.4. Clinical Outcome in the Two Treatment Groups

The two patient groups ASA and SM had a similar risk for long-term all-cause mortality (8.72 vs. 7.84%; *p* = 0.42) short-term all-cause mortality (1.12 vs. 1.27%, *p* = 0.93, Figure 1), CV mortality (2.48 vs. 3.66%; *p* = 0.26) and SCD (1.78 vs. 0.76%; *p* = 0.20, Figure 2a,b). The incidence of post-treatment stroke was also not different between the two groups (0.36 vs. 1.01%; *p* = 0.64, Figure 2c). There was no evidence for publication bias with Egger’s test for any of the clinical outcomes assessed.

### 3.5. Safety Outcomes

ASA was associated with a lower risk of peri-procedural complications (5.57 vs. 10.5%; *p* = 0.04) but a higher risk of re-interventions (10.1 vs. 0.27%; *p* < 0.001) and pacemaker implantation (12.4 vs. 4.31%; *p* = 0.0004, Figure 3) compared to SM. Clinical outcome of the two groups of treatments are summarized in Figure 4. There was no evidence for publication bias with Egger’s test for any of the clinical outcomes assessed.

### 3.6. LV and LA Structure and Function and Clinical Symptoms 

During follow-up, ASA resulted in a lower reduction in LVOT gradient (−47.8 vs. −58.4 mmHg, *p* = 0.01, Figure 5) and less improvement of clinical symptoms compared to SM: angina 53.2 vs. 84.2%, *p* = 0.02 and New York Heart Association (NYHA) class III/IV, 82.4 vs. 94.5%, *p* < 0.001, (mean reduction −1.16 vs. −1.51, *p* = 0.03) (Figure 6a,b and Appendix A) despite a similar reduction in IVSd (−4.07 vs. −4.50 mm, *p* = 0.10), similar increase in LVEDD and similar reduction in EF (*p* = 0.32 and *p* = 0.83, respectively) (Appendix A). SM resulted in a greater reduction in LA size compared to ASA (*p* = 0.04, Appendix A).

### 3.7. Relationship between Demographics and LV Parameters with Peri-Procedural Complications

To test the interaction between demographics and LV structural and functional parameters with peri-procedural complications, we performed a meta-regression analysis. No interaction was found between peri-procedural complications and age, (*p* = 0.58 and *p* = 0.85, respectively) female gender (*p* = 0.24 and *p* = 0.38, respectively) or mean reduction of IVSd (*p* = 0.12 and *p* = 0.68, respectively, Appendix A). Also, no interaction was found between peri-procedural complications and mean reduction of LVOT gradient in the ASA group (*p* = 0.09), but in the SM group, fewer peri-procedural complications were associated with a higher percentage mean reduction of LVOT gradient (*p* < 0.001, Appendix A).

### 3.8. Relationship between Demographics and LV Parameters with Long and Short-Term Mortality

The longer mean follow-up was related to all-cause mortality in the two patient groups, ASA and SM (*p* = 0.02 and *p* = 0.001, respectively), but there was no interaction with the year of publication (*p* = 0.85 and *p* = 0.80, respectively) or age (*p* = 0.81 and *p* = 0.11, respectively, Appendix A). The higher female prevalence was associated with increased all-cause mortality (*p* = 0.01) only in the SM group but not in the ASA group (*p* = 0.287, Appendix A). Also, there was no interaction between mean reduction of LVOT gradient (*p* = 0.80 and *p* = 0.21, respectively) and mean reduction of IVSd (*p* = 0.22 and *p* = 0.65, respectively). Similar results were found in short-term mortality (Appendix A).

### 3.9. Relationship between Demographics and LV Parameters with Clinical Symptoms

The fall in the NYHA class was associated with younger age (ASA: β = −0.654 (−0.041 to 0.194), *p* = 0.005; and SM: β = −0.039 (−0.012 to 0.606), *p* = 0.02] and greater reduction in mean LVOT gradient (ASA: β = −0.139 (−0.368 to 0.091), *p* = 0.004; and SM: β = −0.490 (−0.835 to −0.144), *p* < 0.001). No interaction was found between NYHA class and female gender distribution and the mean reduction of IVSd in the ASA group (*p* = 0.13, *p* = 0.80, respectively), but significant interaction was found in the SM group (*p* = 0.01, for both Appendix A). In addition, no interaction was found between year of publication and NYHA class in the two groups (*p* = 0.38 and *p* = 0.40, respectively Appendix A).

The decline in angina class was also associated with younger age in the two treatment groups (ASA: *p* = 0.001 and SM: *p* = 0.04), less prevalence of female gender (ASA: *p* = 0.04, and SM: *p* = 0.02) and higher percent reduction of IVSd (ASA: *p* = 0.04 and SM; *p* = 0.01). No interaction was found with the mean reduction of LVOT gradient (*p* = 41 and *p* = 0.81, respectively) and year of publication (*p* = 0.71 for ASA), but percent improvement of angina was associated with more recent years of publication (*p* < 0.001) in SM group (Appendix A).

### 3.10. Risk of Bias Assessment

The two included CTs had a low risk of bias (Appendix A). Many of the cohorts are of good quality, and approximately 20% of them are of quality (Appendix A). Also, there was no evidence for publication bias as evaluated by the Egger’s test for our findings.

## 4. Discussion

The findings of meta analyzing the available results of ASA and SM for symptomatic patients with HOCM can be summarized as follows: (1) the ASA patients were slightly older but had similar female gender distribution and LV structure and function parameters despite a smaller left atrial size compared to SM patients; (2) The two procedures carried similar short- and long-term risks for all-cause mortality, cardiovascular mortality, sudden cardiac death and stroke; (3) ASA was associated with a lower risk of peri-procedural complications but a higher risk of re-interventions and pacemaker implantations compared to SM; (4) During follow-up, SM patients had greater reduction in LVOT gradient and greater improvement of clinical symptoms compared to ASA despite similar reductions in IVSd, and LVEF, and similar increases in LVEDD but a greater reduction in LA size compared to ASA; (5) In ASA, there was no interaction between peri-procedural complications and the degree of fall of LVOT gradient, in contrast to SM; (6) Although the longer follow-up period was related to long term all-cause mortality, the interaction with the female gender was only in the SM group; and (7) The fall of NYHA class and improvement of angina was related to young age and the degree of reduction in LVOT gradient irrespective of gender, and to interaction with female gender only in the SM group.

Despite the difference in the nature of the two treatment procedures, the results of this meta-analysis show similar success rates of the two approaches for reducing severe LVOT obstruction in patients with HOCM. Being a surgical procedure, SM was associated with a longer hospital stay compared with ASA, which is catheter-based procedure that usually needs only a couple of nights’ hospital stay. Also, with the aim of inducing acute localized infarct at the top of the interventricular septum where the conduction system runs, using alcohol injection, and with a fear of causing complete heart block, ASA needs pacing until the surrounding myocardial edema subsides and native conduction is well established. This, of course, does not preclude the need for permanent pacing for the two procedures if the need arises. The two procedures had similar survival, all-cause and cardiovascular results, and even stroke was not different between them, in contrast to what old studies showed in patients treated with SM [38,39]. These results are compatible with the recently published long-term data showing the high safety of alcohol septal ablation, with the five years survival rate at 98.9% [40].

The intriguing part of our analysis is the long term related greater reduction of LVOT gradient in the SM group compared to the ASA group, which seems to be associated with better improvement of symptoms compared to the ASA patients. This difference is despite similar changes in LV dimensions and ejection fraction [27,29,30]. With improvements in ASA techniques, and the use of small balloons and the injection of only 1 mL of pure alcohol to induce a small infarct area, investigators reported better procedure outcomes in terms of incidence of heart block and arrhythmias [41]. However, this depends on the anatomical variations of the septal coronary artery branches targeted for the procedure [42]. Even when, in some patients, ASA leaves a small area of upper septal hypertrophy, the remodeling of the segment as well as the resulting right bundle branch block in most patients causes dyssynchronous LVOT and less obstruction [43], with its implications on symptomatic improvement. Conversely, SM entails shaving a significant segment of the upper septal myocardium, thus achieving a wider outflow tract with an expected larger remodeling region as our analysis showed, which also explains the fewer long term associated symptoms we found [22,27,30]. The interaction between symptomatic improvement and degree of fall of LVOT obstruction in females could be explained on the basis of better relief of LVOT resistance in the subset of patients with baseline signs of stiff LV, as shown in the larger left atrium which reduced in size at follow up. 

Our results are almost in line with the previously published meta-analysis, 10 years ago [44], but show more long-term reduction of LVOT gradient in the SM group compared to ASA (94.5 vs. 82.4%) along with better symptomatic improvement. The differences between the two studies could be explained on the basis of no association of the reduction of LVOT gradient with improved NYHA class, the methods used for calculating the fall in NYHA III/IV class between baseline and follow up, in addition to the use of only simple pooled analysis without impact of sample size rather than inverse variance, as reported 10 years ago. Our analysis showed significant correlation between mean reduction of LVOT gradient with the improved NYHA class and more reduction of the mean IVSd and improvement of chest pain. The interaction of young age with the extent of symptomatic improvement in SM, we found, is of particular clinical relevance. Despite such differences, the two procedures have significant pros and cons when it comes to preference and clinical choice, that is, the surgical SM requires bypass circulation and long hospital stays compared with trans-catheter ASA, which is followed by a couple of nights’ hospital recovery. These and other differences should be considered along with other potential risk factors and comorbidities which may impact surgical outcome.

This meta-analysis is based on the available published data from only two clinical trials and the rest were case studies. The analysis was also based on the published data which we trust is accurate. We did not have any hand in the ASA procedure used and the size of the septal infarct created which impacted the degree of LVOT gradient. The difference in the analysis strategy and methods used between our study and that published by Leonardi et al. [44] explains the better symptomatic recovery we found following SM. We would have liked to report on LA volume changes but these data were not available in most analyzed studies. In addition, the long-term mortality by Liebergts et al. was shown to be low and similar in the two treatment groups [45], despite not reporting the clinical symptoms and LVOT reduction. Alcohol septal ablation results in similar short- and long-term risk of mortality, stroke and cardiac complications to those with SM. Better symptomatic recovery at long term follow up might be related to greater regional remodeling of the LV outflow tract following SM, compared with small localized segmental remodeling created by ASA. Based on these results, symptomatic HCM patients with significant LVOTO may be individually recruited for either procedure judged by other factors including age, anatomical evidence for accessible septal coronary artery to the site of segmental obstruction as well as other comorbidities, social and personal preferences.

## 5. Conclusions

Alcohol septal ablation and septal myectomy treatment of HOCM carry a similar risk of mortality. Peri-procedural complications are lower in alcohol ablation but re-intervention and pacemaker dependency are more common. Long-term symptomatic improvement and low LVOT gradient favors septal myectomy. These results might impact the procedure choice in individual patients for the best clinical outcome, with better benefits from septal myectomy in the young compared to alcohol septal ablation in the elderly, who known to have other comorbidities and surgical risks.

## Figures and Tables

**Figure 1 jcm-09-03062-f001:**
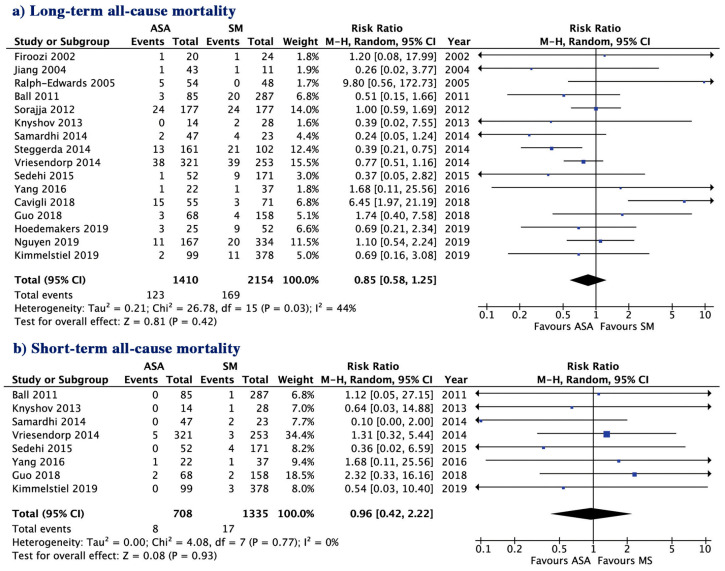
Risk ratio of outcome with alcohol septal ablation (ASA) vs. surgical myectomy (SM); (**a**) Long-term all-cause mortality; (**b**) short-term all-cause mortality. CI, confidence interval.

**Figure 2 jcm-09-03062-f002:**
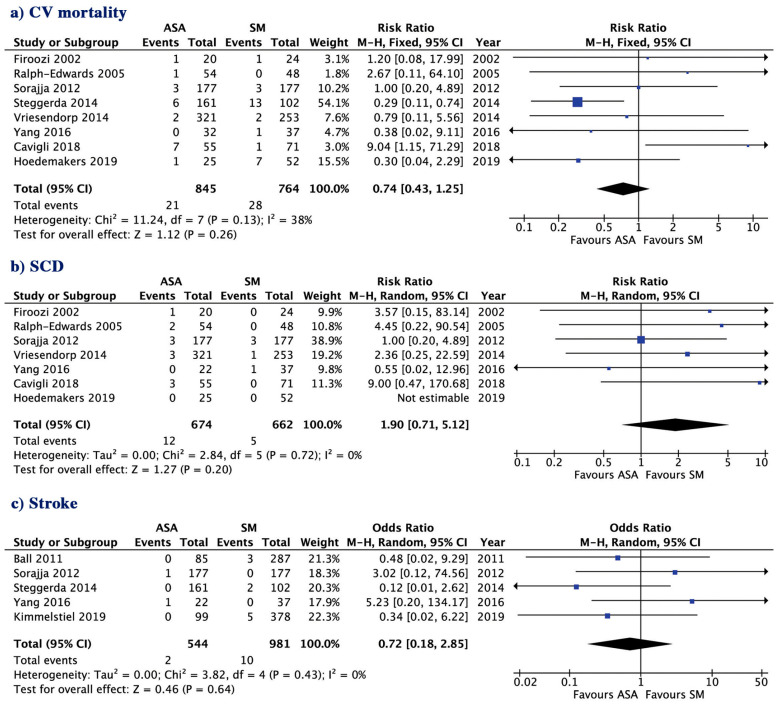
Risk ratio of outcome with septal ablation versus surgical myectomy, (**a**) cardiovascular mortality; (**b**) sudden cardiac death (SCD). ASA, alcohol septal ablation; SM, surgical myectomy; CI, confidence interval. (**c**) Stroke.

**Figure 3 jcm-09-03062-f003:**
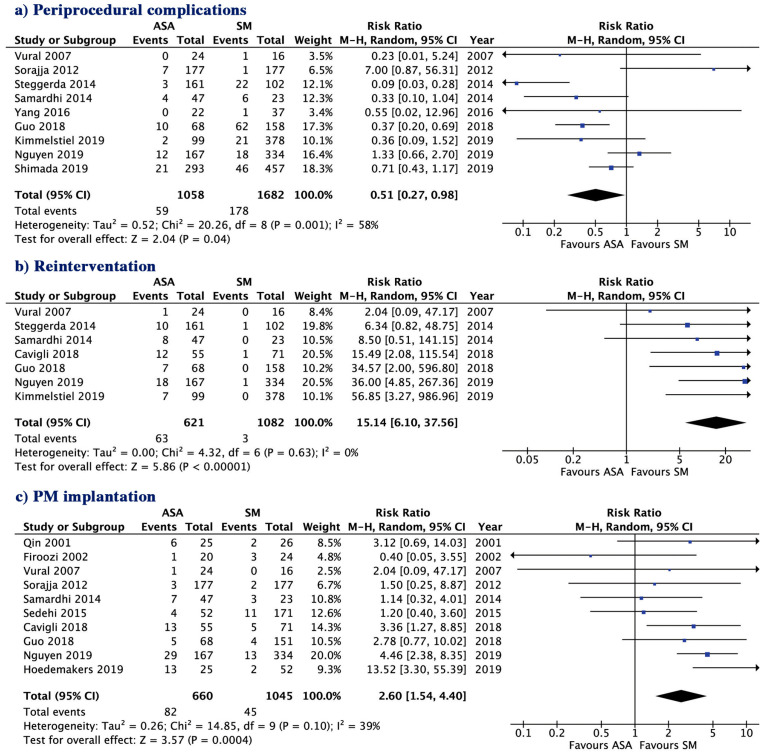
Risk ratio of outcome with septal ablation versus surgical myectomy, (**a**) Peri-procedural complications; (**b**) Re-intervention; (**c**) PM implantation. ASA, alcohol septal ablation; SM, surgical myectomy; CI, confidence interval; PM, pacemaker.

**Figure 4 jcm-09-03062-f004:**
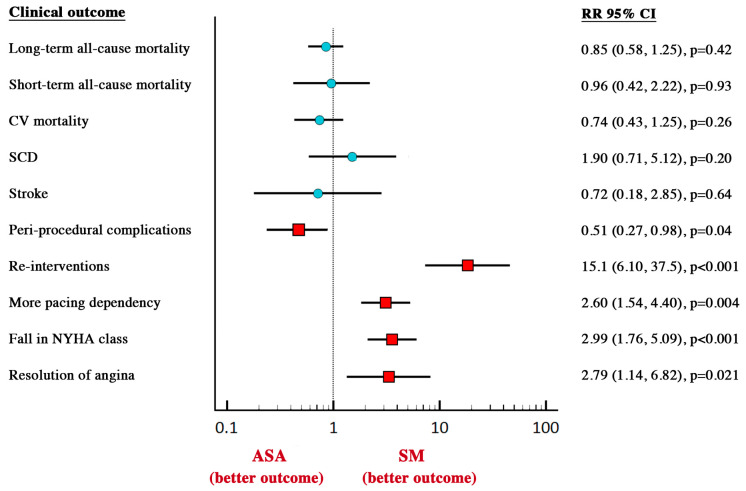
Summary of clinical outcome in two groups of treatment. ASA, alcohol septal ablation; SM, surgical myectomy; CI, confidence interval, CV, cardiovascular, SCD, sudden cardiac death, RR, relative risk, NYHA, New York Heart Association.

**Figure 5 jcm-09-03062-f005:**
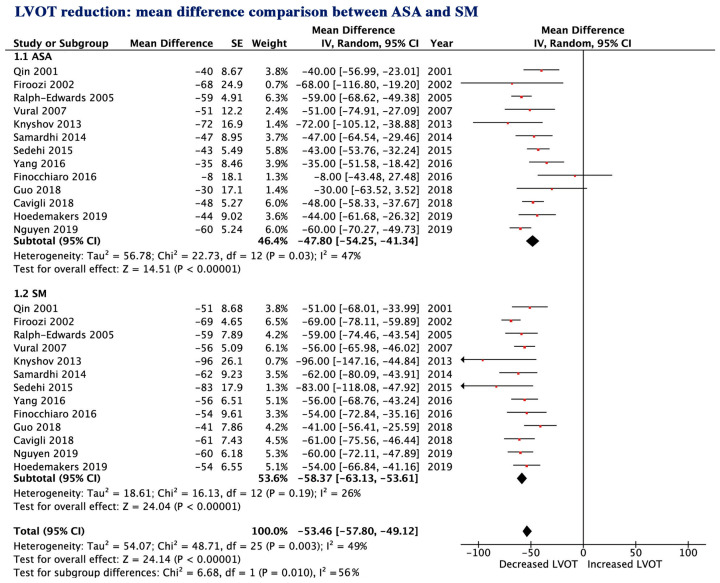
Left ventricular outflow tract (LVOT) gradient mean change; (1.1 ASA) ASA group; ((1.2 SM) SM group. ASA, alcohol septal ablation; SM, surgical myectomy; CI, confidence interval.

**Figure 6 jcm-09-03062-f006:**
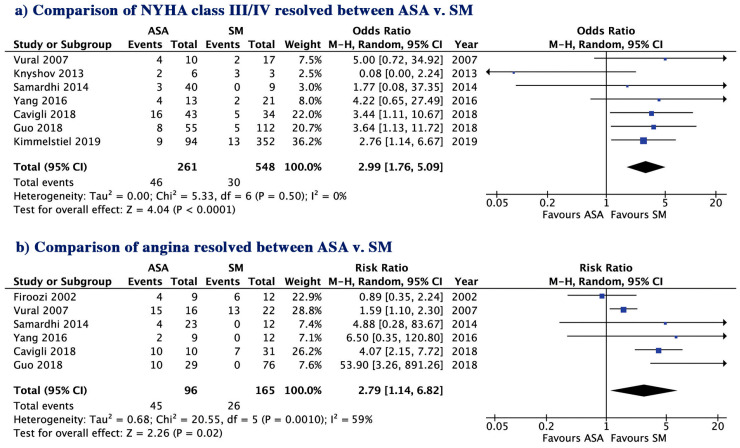
Comparison of clinical improvements between ASA vs. SM; (**a**) NYHA class; (**b**) Angina. ASA, alcohol septal ablation; SM, surgical myectomy; CI, confidence interval. NYHA, New York Heart Association.

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
