# Peer review of "Alcohol Septal Ablation versus Septal Myectomy Treatment of Obstructive Hypertrophic Cardiomyopathy: A Systematic Review and Meta-Analysis"

_jcm, 2020, doi:10.3390/jcm9103062_

Round 1

Reviewer 1 Report

In this meta-analysis, Bytyci et al compare the efficacy of septal myectomy (SM) and alcohol septal ablation (ASA) for obstructive hypertrophic cardiomyopathy (HCM). They included 20 studies – two controlled trials, the rest were cohort studies – of 4547 patients. They found that all cause mortality, CV mortality and stroke did not differ between the two procedures. ASA resulted in less reduction in LVOT gradient and less improvement in symptoms compared to SM. In addition, ASA was associated with lower risk of peri-procedural complications but higher rate of re-intervention and pacemaker dependency.

This meta-analysis is well written with clear methodology and presentation of analysis and results. It remains a timely question clinically with growing numbers of patients being diagnosed with obstructive HCM. The study reports findings similar to a prior meta-analysis conducted 10 years ago by Leonardi et al. However, this analysis adds important information regarding long term reduction in LVOT gradient and symptom improvement favoring septal myectomy. This information is critical in clinical decision making, in particular when considering septal reduction therapy for younger patients. In this meta-analysis, there was an interaction of young age and degree of symptomatic improvement with septal myectomy.

Specific suggestions/queries:

It is intriguing that no difference in pre-procedure degree of mitral regurgitation existed between the two groups. Is there any way to ascertain differences in anatomy that may favor septal myectomy and impact the choices that were made in the cohort studies? i.e. anticipated need for mitral valve repair/intervention, intervention to papillary muscles? Similarly, is there data on follow up echos to assess degree of residual mitral regurgitation between the two groups?

Is there any available data on follow up of biomarkers (NTproBNP) or objective exercise tolerance (stress testing/six minute walk/vo2) to correlate with more subjective findings of NYHA class? This may help in assessing long term outcomes and symptom benefit.

What is the authors’ interpretation of the interaction with female gender in the SM group? Consider addressing this more clearly in the discussion section.

Throughout the manuscript, the Leonardi meta-analysis is cited as reference 43, however, it looks to be reference 42 in the reference section?

Author Response

Reviewer #1

Comments and Suggestions for Authors

In this meta-analysis, Bytyci et al compare the efficacy of septal myectomy (SM) and alcohol septal ablation (ASA) for obstructive hypertrophic cardiomyopathy (HCM). They included 20 studies – two controlled trials, the rest were cohort studies – of 4547 patients. They found that all cause mortality, CV mortality and stroke did not differ between the two procedures. ASA resulted in less reduction in LVOT gradient and less improvement in symptoms compared to SM. In addition, ASA was associated with lower risk of peri-procedural complications but higher rate of re-intervention and pacemaker dependency. This meta-analysis is well written with clear methodology and presentation of analysis and results. It remains a timely question clinically with growing numbers of patients being diagnosed with obstructive HCM. The study reports findings similar to a prior meta-analysis conducted 10 years ago by Leonardi et al. However, this analysis adds important information regarding long term reduction in LVOT gradient and symptom improvement favoring septal myectomy. This information is critical in clinical decision making, in particular when considering septal reduction therapy for younger patients. In this meta-analysis, there was an interaction of young age and degree of symptomatic improvement with septal myectomy.

Specific suggestions/queries:

It is intriguing that no difference in pre-procedure degree of mitral regurgitation existed between the two groups. Is there any way to ascertain differences in anatomy that may favor septal myectomy and impact the choices that were made in the cohort studies? i.e. anticipated need for mitral valve repair/intervention, intervention to papillary muscles? Similarly, is there data on follow up echos to assess degree of residual mitral regurgitation between the two groups?

Response: Thank you for your comment. We have only baseline data (5 papers) that reported MR but not enough data to assess the degree of residual MR or valve replacement during follow up.

Is there any available data on follow up of biomarkers (NTproBNP) or objective exercise tolerance (stress testing/six-minute walk/vo2) to correlate with more subjective findings of NYHA class? This may help in assessing long term outcomes and symptom benefit.

Response: Thank you for your valuable comment. Unfortunately, only two papers reported VO2 consumption so we are unable to undertake a meta-regression.

What is the authors’ interpretation of the interaction with female gender in the SM group? Consider addressing this more clearly in the discussion section.

Response: Thank you for your comment. We have added this point to the discussion section.

Throughout the manuscript, the Leonardi meta-analysis is cited as reference 43, however, it looks to be reference 42 in the reference section?

Response: Thank you for your comment. Reference number has been corrected in the revised manuscript.

Reviewer 2 Report

Bytyci et al submitted their manuscript titled “Alcohol septal ablation versus septal myectomy treatment of obstructive hypertrophic cardiomyopathy: A systematic review and meta-analysis”. The topic the authors chose is interesting and of clinical importance, even though meta-analysis on the same subject are already available. Hypertrophic cardiomyopathy is the most common inherited cardiac disease, with a global distribution and the most frequent cause of sudden cardiac death in the young. The presence of obstruction is related to higher mortality. The authors have performed a good meta-analysis and present with clarity their findings. However there are several weaknesses in the current form of the manuscript.

The authors searched for studies with both invasive therapies for hypertrophic cardiomyopathy, namely alcohol ablation and myectomy. As a result only studies using both treatments were included in the analysis. The reviewer wonders why studies including one method vs placebo were excluded. In the reviewer’s opinion, including studies using either of the methods would improve the solidity of the results. It should be acknowledged that the authors have performed a good analysis of the collected data, even though extensive abbreviations and numbers -in the results sections- are disorienting for the reader and redundant since details are presented in figures/tables.

Another limitation of the manuscript is the discussion section. The implication of the results is poorly interpreted, and important comparisons to other reviews/meta-analyses available are lacking. Instead, the authors chose to present technical details of the two procedures (alcohol ablation or myectomy). As a consequence, at the end of the review, the reader does not clearly understand what is the message/importance delivered by this meta-analysis and to what extend it defers to previous meta-analyses. Another conserving issue is that important papers by experts in the field of HCM are not referenced. Reference [43] is missing.

Author Response

Reviewer #2

Bytyci et al submitted their manuscript titled “Alcohol septal ablation versus septal myectomy treatment of obstructive hypertrophic cardiomyopathy: A systematic review and meta-analysis”. The topic the authors chose is interesting and of clinical importance, even though meta-analysis on the same subject are already available. Hypertrophic cardiomyopathy is the most common inherited cardiac disease, with a global distribution and the most frequent cause of sudden cardiac death in the young. The presence of obstruction is related to higher mortality. The authors have performed a good meta-analysis and present with clarity their findings. However, there are several weaknesses in the current form of the manuscript.

The authors searched for studies with both invasive therapies for hypertrophic cardiomyopathy, namely alcohol ablation and myectomy. As a result, only studies using both treatments were included in the analysis. The reviewer wonders why studies including one method vs placebo were excluded. In the reviewer’s opinion, including studies using either of the methods would improve the solidity of the results. It should be acknowledged that the authors have performed a good analysis of the collected data, even though extensive abbreviations and numbers -in the results sections- are disorienting for the reader and redundant since details are presented in figures/tables.

Response: Thank you for your comment. Our objective was to compare the clinical outcomes between two invasive treatment of HOCM in symptomatic patients despite optimal therapy. This directed us to the need of limiting our analysis to studies which compared the two techniques, as an advanced strategy with respect to comparing individual procedure with placebo.

Thank you for your comment on the abbreviations in the results section. This has now been corrected in the revised manuscript.

Another limitation of the manuscript is the discussion section. The implication of the results is poorly interpreted, and important comparisons to other reviews/meta-analyses available are lacking. Instead, the authors chose to present technical details of the two procedures (alcohol ablation or myectomy). As a consequence, at the end of the review, the reader does not clearly understand what is the message/importance delivered by this meta-analysis and to what extend it defers to previous meta-analyses. Another conserving issue is that important papers by experts in the field of HCM are not referenced. Reference [43] is missing.

Response: Thank you for your comment. This issue in the discussion section has now been clarified.

Round 2

Reviewer 2 Report

Unfortunately, the authors did not modify the discussion section and did not try to improve its quality.

Below are a few suggestions: eg. Lines 227-233, 253-257 need editing. The sentences are very long, with complicating syntax (this phenomenon is repeated throughout the manuscript and needs to be corrected). Lines 247-250 (reply to other reviewer's comment) are inserted without any coherence to the rest of the paragraph meaning and without conveying a clear explanation. Is this explanation a speculation by the authors? 

Other recent meta-analysis are not referenced and discussed at all (eg. Liebregts et al JACC Heart Fail 2015 doi: 10.1016/j.jchf.2015.06.011, Osman et al Clin Cardiol 2019, doi: 10.1002/clc.23113). What is the difference/novelty of this meta-analysis compared to the above?

Figure 5 legend: use same labeling with the rest of the figure for ASA group and SM group (a, b or 1.1, 1.2).

Phrases like "stiff LV", "conduction system runs", "fear of causing complete heart block", "couple of nights hospital recovery", "these and other differences should be considered along with other" etc are examples of informal language that need to be corrected. 

Author Response

Editor’s comments
The manuscript number jcm-914790 entitled: "Alcohol septal ablation versus septal myectomy treatment of obstructive hypertrophic cardiomyopathy: A systematic review and meta-analysis" by Ibadete Bytyçi, Stefano Nistri, Stellen Morner and Michael Henein, has improved with incorporation of the reviewers comments.

The paper provides valuable analysis for clinical management of patients with Hypertrophic Obstructive Cardiomyopathy.

Nevertheless, we believe that the discussion could be improved with the consideration of reviewer#2 comments. In particular with the inclusion of comments on the additional value of the present study compared with the conclusions from Liebregts et al JACC Heart Fail 2015 doi: 10.1016/j.jchf.2015.06.011.

Please also consider the inclusion in the discussion this paper: Batzner A. et al J Am Coll Cardiol. 2018 Dec 18;72(24):3087-3094. doi: 10.1016/j.jacc.2018.09.064, which is the largest single centre experience with alcohol septal ablation.

Response: Thank you for your important comment. We added in the discussion section.
